# Semaphorin 6 Family—An Important Yet Overlooked Group of Signaling Proteins Involved in Cancerogenesis

**DOI:** 10.3390/cancers15235536

**Published:** 2023-11-22

**Authors:** Wiktor Wagner, Błażej Ochman, Waldemar Wagner

**Affiliations:** 1Department of Medical and Molecular Biology, Faculty of Medical Sciences in Zabrze, Medical University of Silesia in Katowice, 41-808 Zabrze, Poland; s86294@365.sum.edu.pl (W.W.); s74817@365.sum.edu.pl (B.O.); 2Laboratory of Cellular Immunology, Institute of Medical Biology, Polish Academy of Sciences, 93-232 Łódź, Poland; 3Department of Hormone Biochemistry, Medical University of Łódź, 90-752 Łódź, Poland

**Keywords:** SEMA6, plexins, RTKs, VEGF, VEGFR1/2, tumor, dendritic cells, therapy prognosis

## Abstract

**Simple Summary:**

The semaphorin 6 family has emerged as a group of proteins for which substantial progress has been made in terms of comprehending the molecular events they engage in and the intricate complexities of their modus operandi within the tumor milieu. Despite the advancements and heightened scholarly attention within cancer research in the realms of targeted therapeutics and immunotherapy, there remains an imperative for continued progress in unraveling the intricacies and pathogenetic mechanisms inherent in tumor progression and in identifying discerning therapeutic targets or diagnostic biomarkers. This comprehensive review provides a nuanced overview and synthesis of the research progress dedicated to elucidating the roles played by members of the semaphorin 6 family in the trajectory of tumor progression, their intricate interplay with relevant signaling pathways, and their impact on immunological phenomena within the tumor microenvironment, positioning the potential clinical relevance of the members of the semaphorin 6 family.

**Abstract:**

According to recent evidence, some groups of semaphorins (SEMAs) have been associated with cancer progression. These proteins are able to modulate the cellular signaling of particular receptor tyrosine kinases (RTKs) via the stimulation of SEMA-specific coreceptors, namely plexins (plexin-A, -B, -C, -D) and neuropilins (Np1, Np2), which share common domains with RTKs, leading to the coactivation of the latter receptors. MET, ERBB2, VEGFR2, PFGFR, and EGFR, among others, represent acknowledged targets of semaphorins that are often associated with tumor progression or poor prognosis. In particular, higher expression of SEMA6 family proteins in cancer cells and stromal cells of the cancer niche is often associated with enhanced tumor angiogenesis, metastasis, and resistance to anticancer therapy. Notably, high SEMA6 expression in malignant tumor cells such as melanoma, pleural mesothelioma, gastric cancer, lung adenocarcinoma, and glioblastoma may serve as a prognostic biomarker of tumor progression. To date, very few studies have focused on the mechanisms of transmembrane SEMA6-driven tumor progression and its underlying interplay with RTKs within the tumor microenvironment. This review presents the growing evidence in the literature on the complex and shaping role of SEMA6 family proteins in cancer responsiveness to environmental stimuli.

## 1. Introduction

Semaphorins (SEMAs) are a large family of extracellular signaling proteins consisting of seven classes, of which classes 1 and 2 are found in invertebrates, and classes 3 to 7 are found in vertebrae [1,2]. They can be secreted, transmembrane, or membrane-bound. They play multiple roles in human physiology, particularly in neurophysiology via axon guidance [3,4], the development of the cardiovascular system [5,6,7], and signaling in the cancer microenvironment, the latter being the main focus of this review. Cellular and molecular processes mediated by semaphorins can occur through various semaphorin signaling modes. Semaphorins encompass three distinct signaling paradigms: juxtacrine, autocrine, or paracrine [8,9]. In the juxtacrine mode, semaphorins are expressed on the cell surface and engage in short-range cell-to-cell interactions with adjacent cells of the same or different types. In the autocrine mode, semaphorins exert their effects on the same cells that produce them, while in the paracrine mode, semaphorins act on nearby cells within the tissue microenvironment [10]. Notably, transmembrane semaphorins are capable of bidirectional signaling, acting in both the “forward” mode and a “reverse” signaling manner. Forward signaling in semaphorins refers to the “classical” signaling pathway, where semaphorins act as ligands and bind to their high-affinity receptors, known as plexins [11]. In the context of semaphorin–plexin interactions, reverse signaling occurs when the cytoplasmic domain of a transmembrane semaphorin interacts with intracellular signaling effectors, resulting in subsequent cellular responses. The intracellular domains of transmembrane semaphorins can interact with various proteins, including kinases and signaling molecules, to modulate cellular processes such as cell migration, adhesion, and cytoskeletal remodeling [12,13,14,15]. While semaphorins have traditionally been recognized for their role as ligands and forward signaling through plexin receptors, reverse signaling adds an additional layer of complexity to their functional repertoire. Furthermore, in addition to their involvement in intercellular signaling (in trans), transmembrane semaphorins and plexins can also form associations on the same cell surface (in cis). This cis association can regulate the functional modulation of other signaling cascades [16,17]. For example, the cis association of SEMA6A with plexin-A4 inhibits the trans interaction of plexin-A4 with SEMA6A expressed by adjacent cells [18]. The main signaling pathways by which semaphorins interact with cells are plexins (A, B, C, and D families) and neuropilins (Np1 and 2, which are needed as coreceptors for class 3 semaphorins) [19,20]. Plexins contain GAP domains (GTPase activating proteins), which can activate and inactivate small GTPases and translate molecular signaling further through serine/threonine kinases (i.e., Raf, GSK3-β, Rho) [21]. SEMA/plexin signaling employs a range of nonreceptor tyrosine kinases, including the FAK, Fer/Fes, Fyn, Pyk2, Src, and Syk families, to govern well-characterized PI3K/AKT and MAPK/ERK downstream intracellular pathways [1,2,22]. An exceptional, important feature of SEMA signaling proteins is the propagation of signals to particular RTKs and tuning of their activity, rendered by their specific counterpart receptors, plexins and neuropilins, which share common domains with RTKs, leading to the coactivation of the latter. MET, ERBB2, VEGFR2, PFGFR, and EGFR, among others, represent acknowledged targets of semaphorins often associated with tumor progression or poor prognosis [22,23]. We examined almost 80 original papers presenting recent achievements in understanding the role of SEMA6s and their cellular counterparts in tumor biology and immune cells present in the tumor environment. This review focuses on the usually overlooked family of SEMA6, its role in cancer development, and possible ways of using this knowledge to diagnose and treat these deadly diseases.

## 2. Semaphorin 6 Family

The semaphorin 6 family contains 4 SEMAs, A, B, C, and D, all of which are transmembrane proteins. Semaphorins categorized within the semaphorin 6 family consist of structural components, namely, the SEMA domain, transmembrane domain, and cytoplasmic domain. The SEMA domain contains binding motifs for plexin and neuropilin receptors, playing a crucial role in receptor interactions and their functional mediation. Notably, the cytoplasmic domains of semaphorin 6 family members are relatively elongated compared to those found in other semaphorin classes [24,25,26]. These semaphorins orchestrate signal transduction through low-affinity interactions, initiated by the activation of the appropriate plexin receptors via the SEMA domain. During this process, the SEMA domain forms a homodimeric face-to-face arrangement, engaging two plexin monomers and adopting a complex seven-bladed β-propeller-fold structure. This intricate architecture ultimately serves as the binding interface for heteromeric semaphorin–plexin complexes on the SEMA domains, culminating in the assembly of heterotetrameric structures [27,28,29,30]. Among the various plexin classes, semaphorin class 6 triggers signal transmission through interactions with class A plexins (plexins A1, A2, A3, and A4) [30,31]. The binding affinity of semaphorin–plexin interactions is modulated by van der Waals forces, guided by shape complementarity between the relevant positions of the interacting molecules, contributing to binding selectivity between semaphorin–plexin pairs. Additionally, it should be noted in the context of the impact of binding affinity on signal transduction that lower affinity may exert a favorable influence on moderate reverse signaling facilitated by semaphorin 6 family members. The structural mechanism underlying plexin-A activation by SEMA6A was unveiled by Nogi et al. [32]. Prior to interaction with plexin-A2, the ectodomain of SEMA6A adopts a face-to-face homodimer configuration, while plexin-A2 assumes a “head-on” homodimer arrangement. Given that the structure of the SEMA6A–plexin-A2 signaling complex revealed a 2:2 heterotetramer configuration, wherein the two plexin-A2 monomers dissociate and bind onto the upper face of the SEMA6A homodimer, it is considered that homodimer-to-heterodimer transitions of cell-surface plexin may constitute the structural mechanism through which the ligand-binding signal is transduced to the cytoplasmic region, inducing GAP domain rearrangements and activation [30,32,33]. In the context of retrograde signal activation during interactions between semaphorin 6 family members and class A plexins, in addition to the influence of binding affinity, avidity, and coreceptor effects, cis–trans interactions may also play an important role [34]. Trans interactions between cell-bound semaphorins and plexins constitute a distinctive mode of action [34,35]. Nevertheless, the significance of cis interactions in the regulation of signal transduction pathways during cis interactions between semaphorin 6 family members and class A plexin receptors has been demonstrated. Investigated cis interactions may regulate cellular processes by either stimulating, as observed in the interaction between SEMA6A and plexin-A2 in the retina, or inhibiting them, as has been demonstrated in the cis interaction between SEMA6B and plexin-A2 during axon developmental processes [17,18]. The cellular effects mediated by the signaling pathways initiated by semaphorin 6 family members are diverse and manifest in various tissues. SEMA6A participates in the migration process of granule cells in the cerebellum through interactions with plexin-A2 [36]. Furthermore, SEMA6A engages with plexin-A2 and Plexin-A4, contributing to the termination of neuronal migration [37]. The interaction of SEMA6B with plexin-A2 is involved in the process of axon navigation [17], and associations have also been demonstrated between the occurrence of specific SEMA6B mutations and the presence of progressive myoclonic epilepsy [38,39]. Both SEMA6C and SEMA6D utilize plexin-A1 to exert their influences on cellular activity [33,40]. The significance of SEMA6C has been demonstrated in the regulation of cellular connections between oocytes and granulosa cells, facilitating proper communication critical for ovarian follicle development [41]. Additionally, it plays a role in neuromuscular communication, where a substantial reduction in SEMA6C concentration has been observed after muscle denervation, with the highest immunoreactivity detected in neuromuscular junctions [42]. In addition to the aforementioned direct interaction of SEMA6D with plexin-A1, SEMA6D also engages in interactions with plexin-A4 [33]. SEMA6D forward signaling holds significant implications for bone homeostasis through its interactions with plexin-A1, triggering receptor expressed on myeloid cells 2 (Trem-2), and the adaptor molecule DNAX-activation protein 12 (Dap12), where it participates in osteoclast development [43]. The SEMA6D interaction along the SEMA6D–plexin-A1–Trem-2–Dap12 axis further promotes the activation of dendritic cells and contributes to cardiac morphogenesis through the interaction of plexin-A1 with two receptor-type tyrosine kinases, Off-track (OTK) and vascular endothelial growth factor receptor 2 (VEGFR2) [13,33,43,44]. The characteristic feature of class 6 semaphorins is their multifaceted action in cancer development and progression (see Table 1).

### 2.1. Semaphorin 6A

Semaphorin 6A is the best-known subclass of semaphorins in the SEMA6 family. Its first documented role was in axon guidance and as an axon growth factor in the development of the central and peripheral nervous systems [72,73,74]. However, ongoing studies and progress in advanced molecular methods have unveiled multiple novel processes in oncogenesis that engage SEMA6 proteins. Angiogenesis and neovascularization are processes by which a tumor gains an uninterrupted flow of nutrients, both of which are crucial for the progression of the disease. The development of new vessels within the tumor niche is stimulated by the interaction of vascular endothelial growth factor (VEGF) with its receptors, vascular endothelial growth factor receptor 1 (VEGFR1) and vascular endothelial growth factor receptor 2 (VEGFR2). Interestingly, VEGF signaling has been found to be impaired by SEMA 6 deficiency [75]. In a study by Segarra et al., silencing of SEMA6A led to significant downregulation of VEGFR2 mRNA and protein expression in primary endothelial cells (HUVECs). Furthermore, a decrease in VEGFR2 levels was found to be critical for endothelial cell survival, and this process could not be compensated by exogenous VEGF. These observations were confirmed in mouse models deprived of SEMA6A. Complementary experiments using SEMA6A-null mice with reduced VEGFR2 expression showed impaired laser-induced choroidal angiogenesis as well as compromised angiogenesis in melanoma and lung tumors in these mice. Finally, disrupted VEGF signaling and angiogenesis translated into a smaller tumor volume compared to mice with normal SEMA6A expression [75]. Collectively, control of VEGFR2 expression via SEMA6A expression impairment may be regarded as a promising new neoadjuvant modality to restrict tumor vascularization. Interestingly, a similar inhibitory effect on the vascularization of tumors in mice was achieved after the treatment of animals with SEMA6A-1 soluble extracellular domain (SEMA-ECD). Matrigel isolated from mice treated with SEMA-ECD at a rate of 1 mg/kg had a significantly lower density of developed blood vessels. Moreover, those vessels were over 2-fold shorter, and their branching points were markedly reduced. These results implied that SEMA-ECD inhibited neovascularization in tumor cells by arresting VEGF-driven Src, FAK, and ERK kinase signaling (Figure 1). This fact is certainly important in the context of therapeutic anticancer modalities and warrants further investigation in the future [76].

In most cancer types, upregulation of SEMA6A levels is associated with an overall poorer prognosis, increased invasiveness, and inhibition of apoptosis of mutated cells. These observations were confirmed in malignant glioma [45], gastric cancer [48], oral carcinoma [52], hepatocellular carcinoma [54,55], renal cell carcinoma [58], and melanoma [64,65,66], where SEMA6A activates pro-survival and pro-proliferative cellular signaling, including the PI3K/AKT, MAPK, and NFkB pathways [64]. In all these examples, the SEMA6A protein could be employed as a potential diagnostic biomarker and/or therapeutic target.

In contrast, low SEMA6A levels correlated with a worse prognosis for particular types of cancers. Such evidence was found for glioblastoma [46] and lung cancer [60]. Upregulation of SEMA6A levels in mice with lung cancer resulted in lower tumor volume. The apoptosis rate of cancer cells was increased, mainly through the FADD-associated apoptosis pathway [77]. In human lung cancer lines, overexpression of SEMA6A resulted in decreased migration due to the activation of the nuclear factor erythroid 2 p45-related factor 2 (NRF2)/heme oxygenase-1 (HMOX) axis, which translated into increased overall survival (OS) and a decreased recurrence rate [60]. In addition, SEMA6A level was found to be higher in nonsmoking lung cancer patients; thus, it could serve as a good biomarker for this narrow group of patients [78]. A similar association with lung cancer among nonsmokers was reported for SEMA5A; however, poor survival among nonsmoking women with NSCLC was related to transcriptional and translational downregulation of SEMA5A in cancer tissue [79].

Finally, there is emerging evidence indicating that SEMA6A can drive the drug resistance of cancer cells through a remodeling of their cytoskeletons. SEMA6A was found to form complexes with beta-actin in the cytoskeleton and to possibly be involved in the modulation of tubulin isotype composition and thus microtubule dynamics in human ovarian cancer cells [80]. Using quantitative PCR, Prislei and colleagues have shown that beta-tubulin III (TUBB3) overexpression/silencing correlates with SEMA6A expression in ovarian A2780 cancer cells. More importantly, the levels of SEMA6A were also higher in drug-resistant and BRAF-mutant melanoma [64]. In their study, Loria and colleagues observed that SEMA6A is partially engaged in the control of actin cytoskeleton remodeling of BRAF-mutated melanoma, which drives their fast rate of proliferation and survival. Moreover, the inhibition of BRAF and MEK kinases by combined dabrafenib + trametinib treatment of BRAF-mutated melanoma led to the activation of the SEMA6A/RhoA/YAP pathway, which resulted in the remodeling of the cytoskeleton and a reduction in targeted therapy efficiency. In particular, YAP is perceived as an oncogenic factor conferring cancer cell stemness, drug resistance, and metastasis associated with cellular cytoskeletal tension and cell-autonomous sensing and responsiveness to the stiffness of the tumor niche [81]. Finally, by unchaining BRAF-mutant melanoma from the stimulatory effects of cancer-associated fibroblasts via the depletion of SEMA6A and its related switched-off pro-survival (PI3K/AKT) and pro-proliferative (MAPK, ΝFκΒ) pathways, the authors were able to rescue the efficiency of BRAF/MEK inhibition against melanoma. In other words, these findings reveal SEMA6A as a new potential targetable protein to reverse drug resistance phenomena in cancer.

### 2.2. Semaphorin 6B

SEMA6B is a transmembrane protein which cooperates with the plexin-A4 receptor and plays a vital role in neuron organization via axon guidance [17,82], the development of hippocampal mossy fibers [83], and axon outgrowth inhibition in lesioned nerve tissue [84]. Moreover, *SEMA6B* gene mutations are a leading factor in the development of SEMA6B-related progressive myoclonic epilepsy (PME-11) [38,39,85,86,87,88,89].

Under pathological conditions, SEMA6B could be involved in thyroid cancer development. SEMA6B levels were found to be upregulated in these malignant tissues [67,68]. While enhanced SEMA6B improved cancer cell viability and invasiveness through the modulation of the NOTCH pathway, silencing of SEMA6B made cancer cells less viable and invasive. Taken together, these results showed SEMA6B is a potential drug target in the prevention and treatment of thyroid gland cancer [67]. Another study aimed at building a prognostic prediction model of thyroid carcinoma employed the pattern of *SEMA6B* gene expression in these malignant tissues [68]. In a *SEMA6B* gene model, upregulation of *SEMA6B* expression resulted in an overall poorer prognosis for patients. The study implied that *SEMA6B* expression, along with other genes tested (*PPBP*, *GCCR*), could be used as a prognostic marker in thyroid carcinoma.

SEMA6B levels were also found to be indicative of prognosis in breast cancer, the most diagnosed cancer among women. A study by Kuznetsova et al. demonstrated an abnormal methylation rate (38%) of the CpG islands in the *SEMA6B* gene, along with a frequent downregulation of SEMA6B expression in 44% of breast tumor samples tested (assayed by real-time PCR) [62]. Contrarily, methylation of the promoter region of *SEMA6B* was not observed in cultured breast cancer-derived cell lines MCF7 and T47D [62]. Interestingly, the potential role of a specific isoform, SEMA6Ba, in mammary tumorigenesis has been proposed; however, more in-depth studies are needed [90]. In contrast to breast cancer, high expression of *SEMA6B* was assessed in other types of tumors: gastric [49], gallbladder [56], and colorectal [69] cancers. Upregulation of *SEMA6B* was frequently associated with increased invasiveness, metastasis, and migration of cancer cells and worse overall prognosis. As expected, silencing the *SEMA6B* gene reversed these negative processes. Li et al. discovered that high *SEMA6B* expression was associated with unfavorable prognosis for patients and adverse polarization of the tumor immunosuppressive microenvironment in colorectal cancer (CRC) patients [69]. These observations suggest that the abatement of SEMA6B expression may be considered an option for cancer treatment; nevertheless, it needs to be further investigated.

Interestingly, plexin-A4, which confers signaling from SEMA6A and SEMA6B, seems to form stable complexes with FGFR1 and VEGFR2 receptors and thus transactivates their signaling pathways [47]. As a result of the intrinsic activity of the SEMA6B–plexin-A4–FGFR1/VEGFR2 axis, endothelial cells (HUVECs) and U87MG glioblastoma cells acquired pro-proliferative stimuli that assembled for pro-angiogenic and pro-tumorigenic signaling in the tumor microenvironment. Indeed, the silencing of plexin-A4 in human umbilical vein endothelial cells resulted in a reduction in proliferation and angiogenesis [47]. Similarly, silencing SEMA6B/plexin-A4 in U87MG glioblastoma cells diminished tumor-forming abilities. In other words, due to reciprocal plexin-A4 coreceptor activity, SEMA6B plays a substantial role in cancer vascularization and responsiveness to environmental stimuli. Based on these observations, the authors of the study suggested that the plexin-A4-SEMA6 axis may be exploited as a target for antitumor therapies [47].

An attractive approach to regulating human *SEMA6B* gene expression was proposed in papers by Collet et al. [91] and Murad et al. [92,93]. These studies investigated the use of specific ligands (agonists) that bind to peroxisome proliferator-activated receptors (PPARs) and 9-cis retinoic acid receptor (RXR). Experiments with the PPARα agonist clofibrate and human glioblastoma T98G cells have shown a strong downregulation of *SEMA6B* gene expression [91]. In follow-up experiments, similar effects were observed in human MCF-7 breast adenocarcinoma cells (the highest expression of *SEMA6B* among cancer cell lines) treated with either fenofibrate (a PPARα activator) or troglitazone (a PPARγ ligand) [92]. Combined treatment of MCF-7 cells with fenofibrate and retinoic acid (RXR agonist) for 72 h substantially decreased SEMA6B protein expression by almost 40%, while in cells exposed to troglitazone and retinoic acid, it decreased SEMA6B protein expression by 70%. The efficacy of this strategy was confirmed in a study on animals aimed at downregulating SEMA6B in rat skeletal muscles [93]. The authors demonstrated the binding of PPARα to the putative PPAR response element (PPRE) in the *SEMA6B* promoter of rat skeletal muscle after fenofibrate treatment. Furthermore, these effects translated into a 2.5-fold lower expression of SEMA6B in the muscles. The concept of using troglitazone or fenofibrate in the control of pro-tumorigenic SEMA6B seems to be an appealing idea since both drugs are already used in clinics to improve the sensitivity and responsiveness of muscles and adipose tissues to insulin or to treat severe hypertriglyceridemia and dyslipidemia. Collectively, these interesting studies emphasized the possibility of pharmacological intervention with fenofibrate against cancer cells via drug repurposing.

### 2.3. Semaphorin 6C

Semaphorin 6C is the least well-known member of the SEMA6 family. It triggers cellular signaling through plexin-A1. Predominantly, widespread expression of SEMA6C has been found in the murine brain, developing embryos, and adult animals, suggesting its important role in neurogenesis and synapse stabilization or formation at postnatal stages [94,95]. Indeed, early functional studies in 2002 focused on the identification of a new axon’s guiding cues delivered new classes of SEMAs, namely SEMA6C and SEMA6D [95]. Substantially, the function of SEMA6C was associated with the inhibition of axonal extension of PC12 cell-differentiated neurons and their outgrowth via directed growth cone collapse in cultured rat hippocampal neurons and rat cortical neurons in a dose-dependent manner. Complementary studies by Svensson et al. [42] on the role of SEMA6C in the peripheral nervous system unveiled the presence of strong immunoreactivity against SEMA6C at the neuromuscular junction of rat anterior tibial and hemidiaphragm muscles. Furthermore, following muscle denervation, the expression of SEMA6C decreased, indicating neuronal downstream regulation of SEMA6C in synapses of motoneurons. This evidence strongly supports the idea of the importance of SEMA6C in neuromuscular communication [42].

Unfortunately, our knowledge of the role of SEMA6C in cancer development is particularly sparse. Nevertheless, recent studies performed on both human subjects and mice revealed that SEMA6C could suppress the proliferation of pancreatic cancer cells [96]. The mechanism underlying these observations was associated with the inhibition of the AKT/GSK3 pathway. The activation of this pathway leads to the accumulation of β-catenin, which promotes the expression of cyclin D1, which in turn drives cell cycle progression and cell division [96]. Thus, under conditions of high SEMA6C expression, semaphorin acts as a suppressor of the AKT/GSK3/β-catenin/cyclin D1 axis and slows the proliferation rate of tumor cells. Importantly, according to bioinformatic analysis, many authors have found that a low level of *SEMA6C* expression is a feature of pancreatic cancer, which is related to a lower survival rate of patients and worse prognosis as compared to patients exhibiting a higher expression level of *SEMA6C*. Accordingly, low expression of *SEMA6C* translated into shorter metastasis-free survival of patients. Recently, Zhou and colleagues [57] employed a bioinformatics approach and harnessed the *SEMA6C* gene expression profile, among 11 other chosen genes, to build a prediction algorithm for the immunological microenvironment of pancreatic tumors. According to the authors’ findings, the proposed method based on the immune-related gene prognostic index (IRGPI) can be employed in survival prediction (OS) in patients with pancreatic adenocarcinoma (PAAD) and molecular classification of PAAD.

### 2.4. Semaphorin 6D in the Immunological Landscape of Tumors

SEMAs and plexin-A1/A4, pairs of cellular ligand–receptor molecules, are noticeably represented on cells of the innate and adaptive immune systems, such as granulocytes, macrophages, dendritic cells (DCs), and CD4^+^ and CD8^+^ lymphocytes [97,98,99]. Their multivariate combination of interactions mediates multiple cell–cell contacts among immune and cancer cells and may confer variable clinical therapy responses within the TME. Indeed, for example, the expression of SEMA4A in the cancer niche of NSCLC or the use of recombinant rSEMA4A improves the response to anti-PD-1 monotherapy and boosts the effector function of tumor-infiltrating CD8^+^ T cells in vitro [99]. On the other hand, SEMA3A secreted by a variety of human tumor cells (lung, breast, prostate, glioblastoma, and colorectal cancer cells) has been shown to inhibit tumor-CD4^+^/CD8^+^ lymphocyte interactions and T-cell activation and proliferation [97].

The preliminary role of plexin-A1 in immune cells was reported by the Ting group in 2003 [100]. In this study, authors emphasized the significant and exclusive presence of plexin-A1 in DCs over other immune cells analyzed, indicating plexin-A1 as a marker to distinguish DCs from other APCs. Moreover, the authors identified major histocompatibility complex (MHC) class II transactivator (CIITA), regarded as the master coactivator of MHC class II genes at the promoter level, as the activator of the *Plxna1* promoter (8-10-fold activation) in murine cells. To further assess the role of plexin-A1 in immune function, the authors employed the interference RNA technique to downregulate the cellular level of plexin-A1. The resulting experimental data demonstrated plexin-A1 as a necessary molecule for optimal priming of T-cell activation by protein- or peptide–antigen-pulsed DCs (Figure 2A).

The natural counterparts for SEMA6D-positive CD4^+^ cells are plexin-A1-bearing DCs, which also coexpress plexin-A4, SEMA3A and Np1 receptors. Thus, such reciprocal T-cell–DC cell–cell interactions via Sema6D–plexin-A1 contacts are expected. According to Takegahara and colleagues [43], the incubation of dendritic cells with recombinant soluble SEMA6D induced IL-12 production and upregulation of MHC class II expression. Complementary experiments using plexin-A1^−^/^−^ deficient dendritic cells in the presence of SEMA6D significantly restricted SEMA6D binding to DCs and reduced IL-12 production and MHC class II upregulation. Moreover, cytokine production (IL-2, IL-4, IFNγ) and proliferation by CD4^+^ cells prepared from draining murine lymph nodes were considerably reduced as a result of compromised sensing of environmental stimuli by dendritic cells derived from plexin-A1^−^/^−^ mice (Figure 2B).

Recent evidence on the role of plexin-A1 in DC-mediated T-cell activation addressed the possible molecular mechanism involving the activation of Rho GTPases (known to regulate the actin cytoskeleton) [101]. A study by Eun et al. documented that plexin-A1 localizes to the DC membrane and focal adhesions during DC-T-cell contacts following antigen pulse. Furthermore, coupled DC-T-cell conjugates demonstrated F-actin accumulation at the DC-T-cell interface in DCs but not plexin-A1 shRNA DCs. Following antigen pulsing (ovoalbumin, OVA), DC–T-cell pairs triggered an efficient activation of Rho in DCs. Complementary experiments using plexin-A1-deprived DCs displayed over 50% lower F-actin polarization in DCs, which is vital for the creation of immunological synapses during T-cell–DC interactions [102]. Additionally, OVA-pulsed plexin-A1 shRNA-treated DCs exhibited a lower activation of Rho (interestingly, not of Rac or Cdc42) than control shRNA-treated DCs. Finally, pretreatment of DCs with the Rho inhibitor C3 greatly reduced the accumulation of F-actin in DCs at the interface with T-cells. Collectively, the present study underlines the regulatory role of the plexin-A1–Rho axis in cytoskeletal rearrangements and highlights the major role of this mechanism during the creation of immunological synapses and DC–T-cell interactions (Figure 2C).

Interestingly, semaphorin 6D (possibly paired with plexin-A1) has also been attributed to the generation of immunological memory via the activation of the late phases of the T-cell immune response [103,104]. According to O’Connor et al., the activation of CD4^+^ T cells enhanced SEMA6D expression in vivo and vice versa [103]. As expected, targeting SEMA6D with specific anti-SEMA6D antibodies diminished endogenous T-cell signaling mediated by phosphorylation of the linker of activated T-cells (pLAT; possibly also c-Abl) and accounted for the reduction in the late phase of T-cell activation (Figure 2D).

Collectively, the presented experimental results describe plexin-A1 as a functional receptor for SEMA6D in the immune landscape and suggest the participation of the SEMA6D–plexin-A1 pairing in the early phases of immune reactions via T-cell–DC contacts. Importantly, activated CD4^+^ T-cells can increase antitumor immunity by promoting pro-inflammatory cross-presenting dendritic cells (DCs) and thus strengthen the antitumor effector functions of CD8^+^ [105].

Apart from plexin-A1–SEMA6D interactions, plexin-A4 may also act as a receptor for SEMA6D, as both molecules exhibit widespread expression on the surface of interacting immune cells, including T cells, dendritic cells (DCs), and macrophages [106]. Kang et al. investigated the role of SEMA6D and plexin-A4 in the polarization of macrophages toward the anti-inflammatory subtype [107]. Notably, these regulatory mechanisms employ SEMA6D to function as a receptor itself via reverse signaling [13]. Macrophages can be classified into two distinct subtypes, pro-inflammatory and anti-inflammatory, distinguished by their secretion of various cytokines and mediators and by different factors that drive their polarization into the respective subtype [108]. The process of macrophage polarization into the appropriate subtype is also subject to regulation by alterations in specific metabolic pathways. For instance, in the case of anti-inflammatory macrophages, IL4-induced upregulation of PPARγ and activation of mTOR kinase led to modifications in fatty acid metabolism, which are vital for the polarization process [109,110]. Kang et al. revealed that the expression of SEMA6D affects the direction of macrophage polarization, and its absence disrupts the immune response. They emphasized the significance of the mTOR–SEMA6D–PPARγ axis in linking immunity and metabolism during macrophage polarization (see Figure 3). This axis is under the regulation of mTOR kinase, which orchestrates the expression of SEMA6D. In turn, SEMA6D expression affects PPARγ expression and metabolic processes associated with fatty acid uptake and lipid metabolism remodeling. The lack of SEMA6D leads to impaired polarization of macrophages toward an anti-inflammatory state and heightened pro-inflammatory polarization. In this context, SEMA6D functions as a receptor in reverse signaling through its cytoplasmic domain and the c-Abl kinase to regulate PPARγ expression. The interaction between SEMA6D and its ligand plexin-A4 stimulates reverse signaling, thereby contributing to the polarization of macrophages into the anti-inflammatory subtype [107]. The process of polarizing anti-inflammatory macrophages, in which SEMA6D plays a crucial role, as demonstrated both in vitro and in vivo, holds potential implications for understanding the function of tumor-associated macrophages (TAMs). Manipulating the mTOR–SEMA6D–PPARγ axis, which connects macrophage immunological function and metabolism, offers promise in modulating immune responses and therapeutically influencing macrophage polarization not only in cancer but also in various other diseases. However, the precise role of SEMA6D in macrophage polarization within the tumor microenvironment remains incompletely elucidated.

Beyond the immune system, SEMA6D is involved in the development of the heart by promoting the proliferation of cardiomyocytes [111], preliminary stages of atrioventricular cushion mesenchyme development [112], and proper myocardium organization by enhancing the migration of cardiac cells into trabeculae [13]. Conversely, the pathological proliferation of cells accompanying liver fibrosis has been associated with a higher level of SEMA6D in the blood of patients with chronic hepatitis C (CHC) [113]. Furthermore, SEMA protein levels (SEMA3C, SEMA5A, and SEMA6D) positively correlated with fibrosis stage in CHC. Indeed, antiviral therapy against HCV improved the stage of liver fibrosis and decreased serum SEMA3C/SEMA6D levels. Interestingly, the authors of the study concluded that SEMA6D may serve as a sensitive marker of liver injury in CHC (superior to APRI and FIB-4 in predicting the development of cirrhosis) [113].

### 2.5. Semaphorin 6D in Cancer Tissues

The literature on cancer biology delivers numerous pieces of evidence on the role of SEMA6D in the development of various types of cancers, including osteosarcoma, CRC, lung, renal, breast, pleural mesothelioma, and gastric cancers [50,51,53,59,63,70,71,114,115,116,117,118]. Accordingly, SEMA4D and SEMA6D have been identified as proto-oncogenes in human osteosarcoma cells, and their expression was found to be highly elevated in comparison to normal human osteoblasts [70]. Overexpression of SEMA4D and SEMA6D enhanced intracellular signaling following activation of the PI3K-AKT-mTOR and MAPK pathways through the SEMA4D/SEMA6D coreceptors MET or ERBB2 and VEGFR2. Therefore, activated HOS, MG63, and SaOS2 osteosarcoma cells exhibited markedly increased xenograft formation, colony formation, and proliferation. These observations have been confirmed by Dong and Qu [114], who emphasized the role of circular RNAs circUBAP2 and SEMA6D in the chemoresistance of osteosarcoma patients. Collectively, the study demonstrated that high expression of circUBAP2 and SEMA6D promotes the Wnt/β-catenin signaling pathway and cisplatin resistance, while the microRNA miR-506-3p negatively regulates circUBAP2 and SEMA6D and restrains osteosarcoma invasiveness. In other words, upregulated SEMA6D levels in cisplatin-resistant osteosarcoma cells may be used as a prediction tool for personalized therapy and considered a promising target for the molecular targeted therapy of chemoresistant osteosarcoma [114].

Experimental evidence on the role of SEMAD in the development of breast cancer varies considerably across the available but limited studies [63,115,116]. Gunyuz et al. showed several cell-specific effects of SEMA6D on breast cancer cell behavior that may translate into different clinical outcomes in individual breast cancer patients [115]. While normal MCF10A breast cells with overexpressed SEMA6D increased their proliferation, the opposite effects were reported for malignant MCF7 breast cells (diminished proliferation). In addition, prominent characteristics of cancer cells associated with an enhanced epithelial–mesenchymal transition (EMT) phenotype and cancer aggressiveness advanced in response to *SEMA6D* overexpression in cancer breast cells (increased migration and invasion ability) and in normal breast cells (increased migration). In contrast, the tumorigenic potential of MCF7 and MDA MB 231 cells overexpressing SEMA6D did not translate into a higher ability to create anchorage-independent growth and thus set a new cancer niche. Interestingly, in another study by Chen and colleagues, aimed at the analysis of the gene interaction network in patients with breast invasive carcinoma, several genes characteristic of EMT were elevated two- to threefold (*MITF*, *TCF4*, *SNAI2*, *ZEB2*, *ZEB1*, *GNG11*) in high-*SEMA6D* expression patients, while another key tumor metastatic promoter, matrix metallopeptidase 9 (MMP-9), was dramatically reduced [116]. Positive correlations between *SEMA6D* and *VEGF* genes were also found. Importantly, the correlation of *SEMA6D* expression levels with overall survival in triple-negative breast cancer patients (TNBC) was strongly suggested. The possible role of SEMA6D in the development of breast cancer chemoresistance was raised by Baxter and colleagues [63]. Indeed, transcriptomic data analysis within the estrogen receptor-negative and HER2-positive breast cancer patient subgroups showed significant predictive value of low *SEMA6D* expression among patients susceptible to early recurrence following completed chemotherapy. Furthermore, complementary experiments in vitro demonstrated that reduced levels of SEMA6D through negative control by miR-195 could raise chemoresistance in estrogen-positive breast MCF7 and MDA-MB-175 cancer cells [63]. Collectively, the presented reports indicate a possible modulatory role of SEMA6D in restricting the proliferation rate and chemoresistance of cancer breast cells. In other words, according to in vitro studies and clinical observation, deficiency of SEMA6B (described earlier) and SEMA6D expression are found in breast tumors and coincide with cancer growth, chemoresistance, and recurrence.

In gastric cancer (GC), elevated SEMA6D levels along with the EMT marker Snail were found to be associated with increased invasion, lymph node metastasis, and differentiation of gastric cancer cells [50,51]. Moreover, SEMA6D coupled with plexin-A1 was found to be involved in the neovascularization of gastric tumors [117]. According to Lu and colleagues, the SEMA6D–plexin-A1 complex was highly expressed in vascular endothelial cells within gastric cancer and positively correlated with VEGFR2. Such a close association of plexin-A1 and VEGFR2 was also previously documented in malignant pleural mesothelioma (MPM) [71]. According to the study of Catalano and colleagues, continuous exposure of normal mesothelial cells to asbestos fibers drove excessive expression of SEMA6D and its receptor plexin-A1, which protected met-5A cells from asbestos-induced cytotoxicity. Interestingly, after malignant transformation, pleural mesothelioma (H2052 cells) retained higher expression of SEMA6D and acquired plexin-A1–mediated anchorage-independent survival of MPM cells. In a series of experiments with soluble SEMA6D (SEMA6D-conditioned medium; SEMA6D-CM), the authors showed that SEMA6D-CM increased VEGF-stimulated tyrosine phosphorylation of VEGFR2 in a plexin-A1-dependent manner. Finally, complementary experiments indicated that malignant met-5A cells escape apoptosis by activating the plexin-A1/VEGFR2 pathway and downstream induction of pro-survival NF-κB signaling. Taken together, the study emphasized the crucial role of SEMA6D/plexin-A1 signaling activity in the survival pathway of human mesothelioma cells and underlined a possible molecular target for therapeutic modalities.

Unlike in gastric cancer, the negative control of SEMA6D over VEGF expression was confirmed in CRC, as reported in the study of Lee et al. [118]. In these experiments, the authors abolished the antiangiogenic effect of SEMA6D by IL-17C-driven induction of miR-23a-3p expression, which in turn suppressed the target *SEMA6D* gene. In contrast, miR-1, which is considered a tumor suppressor in prostate cancer and head and neck squamous-cell carcinoma, was downregulated in esophageal cancer in comparison to normal tissue [53]. The authors identified 13 genes targeted by miR-1, among which the *SEMA6D* gene was positively regulated by miR-1 in pathological tissues (twofold increase). Taken together, these reports highlight the substantial role of SEMA6D in the tumor microenvironment and suggest the SEMA6D/VEGF(R) axis as a possible target for pharmacological intervention.

As in the case of breast cancer, SEMA6D seems to possess a multifaceted nature during the development of clear cell renal carcinoma (ccRCC) [59]. Although *SEMA6D* expression was markedly lower in ccRCC tissue samples than in normal tissue, relatively higher *SEMA6D* expression levels in patients with ccRCC correlated with longer disease-free survival. Indeed, patients with an advanced stage of ccRCC (stage IV) presented the lowest level of *SEMA6D* expression. Moreover, the levels of urine SEMA6D in healthy individuals and ccRCC patients reflected *SEMA6D* expression levels in the respective tissues; thus, urine sampling may be harnessed for routine patient testing for ccRCC. Indeed, complementary in vitro experiments documented that overexpression of SEMA6D confers inhibitory effects on the proliferation, migration, and invasion of ccRCC cells. In terms of these results, the authors concluded that SEMA6D may serve as a reliable biomarker to track ccRCC development.

Finally, an in-depth bioinformatic analysis revealed the downregulation of *SEMA6D* and its close association with the development of lung adenocarcinoma (LUAD) [61]. In this study, the *SEMA6D* gene, among others (*CENPF*, *TP53*, and *NRXN1*), was recommended as a prognostic biomarker associated with LUAD. In other words, *SEMA6D* could serve as a tumor suppressor gene in LUAD, and patients with low *SEMA6D* expression are prone to disease progression and shorter OS.

## 3. Concluding Remarks

Semaphorins were first discovered in the developing nervous system, and their role was attributed to axon guidance. However, further studies revealed that transmembrane proteins with the SEMA domain exhibit several pleiotropic activities associated not only with physiological roles (development of the cardiovascular, skeletal, and immune systems) but also with numerous pathologies, including cancer. The incidence of cancer in the human population is constantly rising; fortunately, our knowledge of the complexity of tumor biology is also growing. Thus, progress in medicine and biology in terms of understanding the complexity of SEMAs’ biology may shed a new light on cancer behavior. The investigation of the role of semaphorins encounters substantial challenges due to the complexity of their signaling mechanisms. Members of the semaphorin 6 family demonstrate potential as prospective targets for therapy, offering avenues to enhance the outcomes of oncological treatment by modulating pivotal signaling pathways implicated in metastasis and tumor progression and involved in the angiogenesis, proliferation, migration, and invasion of cancer cells, as well as in shaping the immunological properties of the tumor microenvironment. Furthermore, the observed association between an elevated expression of the semaphorin 6 family and drug resistance, as demonstrated in osteosarcoma and melanoma, should also be noted. Further research into the inhibitory effects of the expression of semaphorin 6 family members and its influence on resistance to anticancer therapies is undoubtedly a crucial pursuit. Achieving this objective will provide a more nuanced understanding of the potential therapeutic implications associated with targeting semaphorin 6 family members expression in chemoresistant tumors. However, the current state of knowledge regarding the precise effects of these interactions and their impact across signaling cascades remains incompletely elucidated. The absence of comprehensive investigations assessing the occurrence of reserve signaling and the implications of cis and trans interactions between tumor cells themselves and those within the tumor microenvironment, such as immune cells and fibroblasts, alongside potential yet undiscovered interactions of semaphorin 6 family members with class A plexins, poses challenges in accurately predicting the direct influence of inhibiting semaphorin 6 family pathways on tumor cells and their potential actions in other tissues. Further investigations should delve into a detailed understanding of factors regulating the expression of semaphorin 6 family members, such as mutations accompanying distinct tumor subtypes, as well as interactions with immune cells within the tumor microenvironment. In conclusion, semaphorins, including the semaphorin 6 family, assume diverse and intricate roles in cancer initiation and progression. Their involvement in multiple signaling pathways and the context-dependent nature of their effects present challenges in comprehending their precise functionalities. Nonetheless, the identification of semaphorins as potential therapeutic targets and diagnostic markers opens new prospects for the development of innovative cancer treatments and the enhancement of patient outcomes. Understanding semaphorins’ function in the field of their substantial role in the tumor microenvironment would favor humankind in the deadly fight against cancer.

## Figures and Tables

**Figure 1 cancers-15-05536-f001:**
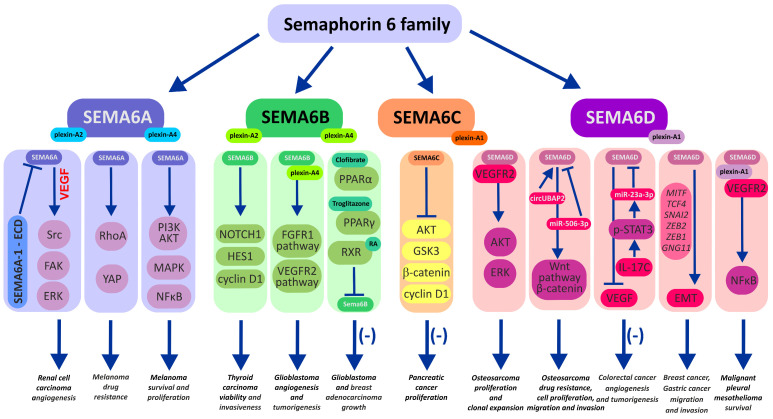
Illustration of semaphorin 6A–D signaling pathways and their interaction with the cellular molecular network in cancer cells. Activation of semaphorin signaling either via the “forward” (via plexins) or “reverse” mode by engaging RTK signaling is often associated with enhanced tumor angiogenesis, metastasis, and resistance to anticancer therapy. Pharmacological intervention (clofibrate, troglitazone, retinoic acid (RA)), molecular downregulation of SEMA6 activity (by dominant negative SEMA6A-1 soluble extracellular domain, SEMA-ECD), downregulation of protein expression (miRNA), or even upregulation of SEMA6 expression (SEMA6C) may be beneficial as adjuvants along with anticancer therapeutics. VEGF: vascular endothelial growth factor; VEGFR1/2: vascular endothelial growth factor receptor 1/2; Src: Src kinase; FAK: focal adhesion kinase; ERK: extracellular signal-regulated kinase; RhoA: transforming protein RhoA; YAP: Yes-associated protein; PI3K: phosphoinositide 3-kinase; AKT: protein kinase B; MAPK: mitogen-activated protein kinase; NfkB: nuclear factor kappa-light-chain-enhancer of activated B cells; NOTCH1: neurogenic locus notch homolog protein 1; HES1: transcription factor HES1; FGFR1: fibroblast growth factor receptor 1; PPARα: peroxisome proliferator-activated receptor α; PPARγ: peroxisome proliferator-activated receptor γ; RXR: retinoid X receptor; GSK3: glycogen synthase kinase 3; Wnt: Wnt signaling pathway; STAT3: signal transducer and activator of transcription 3; EMT: epithelial to mesenchymal transition.

**Figure 2 cancers-15-05536-f002:**
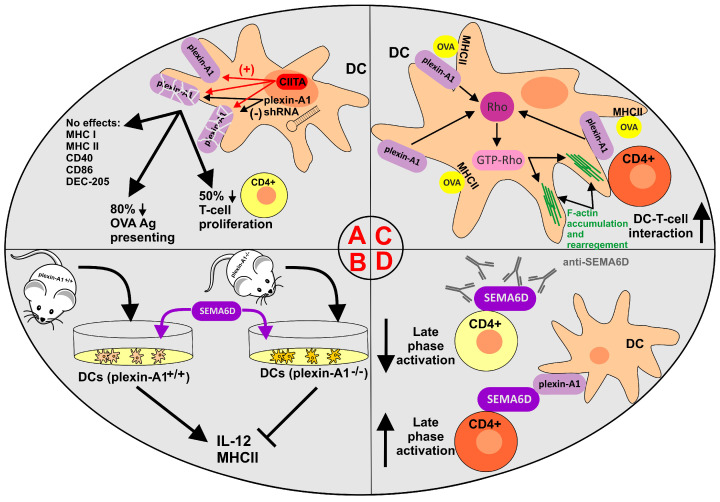
Role of plexin-A1 and SEMA6D components in various immunological processes involving dendritic cells and T cells. (**A**) CIITA activates the plexin-A1 promoter in murine DCs. Stably transfected DCs with plexin-A1 shRNA exhibit compromised OVA antigen presentation and T-cell activation. (**B**) Plexin-A1-deficient DCs isolated from plexin-A1^−^/^−^ mice exhibit compromised IL-12 production and MHC II expression. (**C**) Schematic representation of the role of plexin-A1 in controlling the priming of DC–T-cell interactions. Plexin-A1 is dispensable for Rho and F-actin accumulation and polarization in the immunological synapse of DCs preceding dendrite formation and T-cells adhesion. (**D**) Targeting of Sema6D–plexin-A1 pairing in cocultured OVA-triggered DCs and CD4^+^ cells by anti-SEMA6D antibodies reduced the proliferation rate of T-cells and diminished their activation rate in the late phase of activation (4–6 days). (+)—stimulation; (-) or ⟙—inhibition; **↑**—upregulation; ↓—downregulation.

**Figure 3 cancers-15-05536-f003:**
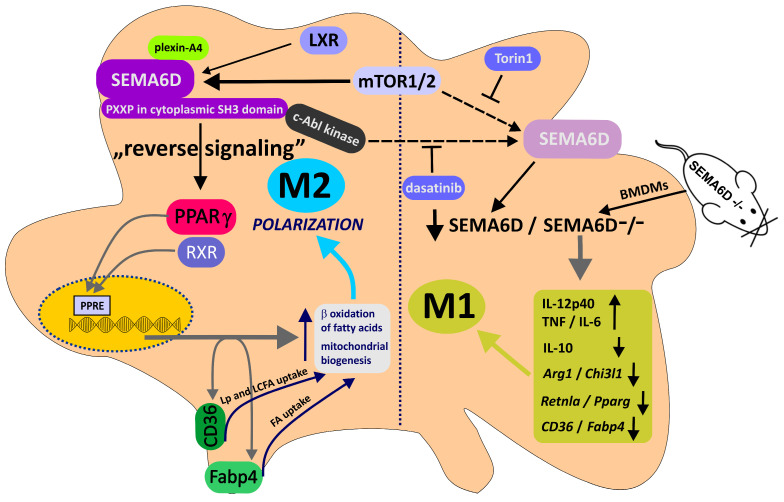
Illustration of the mTOR–Sema6D–PPARγ signaling pathway in controlling macrophage polarization toward anti-inflammatory (antitumorigenic) phenotype M2 via regulation of fatty acid uptake and metabolic reprogramming. mTOR is a critical molecule that controls SEMAD6 expression and regulates the metabolic status of the cell to promote polarization of macrophages. Association of c-Abl tyrosine kinase with SEMA6D with the PXXP region of the cytoplasmic SH3 domain promotes reverse signaling, enhanced by binding plexin-A4. As a result of SEMA6D-driven reverse signaling, PPARγ expression increases and primes metabolic reprogramming events such as fatty acid biosynthesis and fatty acid uptake pathways (CD36, Fabp4). SEMA6D–PPARγ signaling is indispensable for metabolic reprogramming. Abnormalities in this pathway account for impaired fatty acid uptake and metabolic reprogramming, leading to aberrant macrophage polarization. Inhibition of upstream mTOR1/2 with Torin1 or activity of c-Abl tyrosine kinase with dasatinib suppresses the expression of PPARγ and the anti-inflammatory signature genes *Arg1* and *Chi3l1*. Similarly, mutation of the SH3 domain in SEMA6D or absence of SEMA6D accounts for defective anti-inflammatory macrophage polarization (low expression of anti-inflammatory signature genes: *Retnla*, *Il10*, and receptors *Cd36*, *Fabp4*) along with exaggerated inflammatory responses (IL-12p40, TNF, IL-6). mTOR1/2: mammalian target of rapamycin 1/2; PPARγ: peroxisome proliferator-activated receptor gamma; RXR: retinoid X receptor; PPRE: peroxisome proliferator hormone response elements; LXRα: liver X receptor alpha; Arg1: arginase 1; Chi3l1: chitinase 3 like-1; Retnla: resistin-like molecule alpha; CD36: receptor for lipoproteins; Fabp4: Lp: lipoproteins; FA: fatty acids; LCFA: long-chain fatty acids; BMDMs: bone marrow-derived macrophages. ⟙—inhibition; **↑**—upregulation; ↓—downregulation.

**Table 1 cancers-15-05536-t001:** The pattern of semaphorin 6 expression in tumors and cancer cells in vitro.

SEMA6A		SEMA6B		SEMA6C		SEMA6D	
**↑**	**↓**	**↑**	**↓**	**↑**	**↓**	**↑**	**↓**
Malignant glioma [45]	Glioblastoma [46]	U87MG glioblastoma cells [47]	—	—	—	—	—
Gastric cancer [48]	—	Gastric cancer [49]	—	—	—	Gastric cancer [50,51]	
Oral carcinoma [52]	—	—	—	—	—	Esophageal cancer [53]	—
Hepatocellular carcinoma [54,55]	—	Gallbladder cancer [56]	—	—	Pancreatic cancer [57]	—	—
Clear cell renal carcinoma [58]	—	—	—	—	—	—	Clear cell renal carcinoma [59]
—	Lung cancer [60]	—	—	—	—	—	Lung adenocarcinoma [61]
	—	—	Breast cancer [62]	—	—	—	MCF7 breast cancer cells [63]
Melanoma [64,65,66]	—	—	—	—	—	—	—
—	—	Thyroid carcinoma [67,68]	—	—	—	—	—
—	—	Colorectal cancer [69]	—	—	—	Osteosarcoma [70]	Pleural mesothelioma [71]

↑—increased expression; ↓—decreased expression.

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
