# Peer review of "Semaphorin 6 Family—An Important Yet Overlooked Group of Signaling Proteins Involved in Cancerogenesis"

_cancers, 2023, doi:10.3390/cancers15235536_

Round 1

Reviewer 1 Report (Previous Reviewer 2)

Comments and Suggestions for Authors

This revised review has satisfactorily addressed my comments. Some sections have been reorganized to enhance its presentation. This review will be a valuable addition to literature. I support its publication.

Author Response

Reviewer 1: This revised review has satisfactorily addressed my comments. Some sections have been reorganized to enhance its presentation. This review will be a valuable addition to literature. I support its publication.

Response to Reviewer 1. Thank you very much for your kind words and careful review of our paper.

Reviewer 2 Report (New Reviewer)

Comments and Suggestions for Authors

The manuscript has drammatically improved in the current version. 

Further comments

1 - Please go through the Simple Summary, it is currently difficult to follow.

2 - Please comment on the text about the possibilities of application of the knowledge about Semaphorin 6 protein and cancer. How would the authors think Semaphorins can be used in cancer diagnosis/treatment? Please include this information on the text.

Comments on the Quality of English Language

The English is overall fine, but Simple Summary needs checking.

Author Response

Reviewer 2: The manuscript has drammatically improved in the current version. Further comments:

Comment 1 and 3. Please go through the Simple Summary, it is currently difficult to follow. The English is overall fine, but Simple Summary needs checking.

Response to Comment 1 and 3. Thank you very much for your kind words and careful review of our paper. We have rephrased Simple Summary to make it clear for readers and paragraph has been checked by a colleague fluent in English writing followed by English editing with American Journal Experts Digital Editing tool according to your suggestions. See text lines: 11-21, highlighted in yellow.

Comment 2. Please comment on the text about the possibilities of application of the knowledge about Semaphorin 6 protein and cancer. How would the authors think Semaphorins can be used in cancer diagnosis/treatment? Please include this information on the text.

Response to Comment 2. Thank you for pointing to this issue. Indeed, semaphorin 6 family members demonstrate potential as prospective targets for therapy, offering avenues to enhance the outcomes of oncological treatment by modulating pivotal signaling pathways implicated in metastasis and tumor progression. On the other hand, detailed SEMA6 interactions within tumor niche and their wide impact across signaling cascades remains incompletely elucidated and poses challenges in accurately predicting the direct influence of inhibiting semaphorin 6 family pathways on tumor cells and their potential actions in other tissues. Nonetheless, the identification of semaphorins as potential therapeutic targets and diagnostic markers opens new prospects for the development of innovative cancer treatments and the enhancement of patient outcomes. Understanding semaphorins’ function in the field of their substantial role in the tumor microenvironment would favor humankind in the deadly fight against cancer. These considerations have been discussed in manuscript, in section Concluding remarks, see text lines 602-632.

The possible clinical application of the scientific knowledge about the role of SEMAs in cancer development has also been discussed within particular paragraphs describing involvement of SEMA subclasses in cancer development; see text lines shown below:

Lines 243-245 - In other words, these findings reveal SEMA6A as a new potential targetable protein to reverse drug resistance phenomena in cancer.

Lines 261-262 - The study implied that SEMA6B expression, along with other genes tested (PPBP, GCCR), could be used as a prognostic marker in thyroid carcinoma.

Lines 312-314 - The concept of using troglitazone or fenofibrate in the control of protumorigenic SEMA6B seems to be an appealing idea since both drugs are already used in clinics to improve the sensitivity and responsiveness of muscles and adipose tissues to insulin or to treat severe hypertriglyceridemia and dyslipidemia. Collectively, these interesting studies emphasized the possibility of pharmacological intervention with fenofibrate against cancer cells via drug repurposing.

Lines 347-350 - Recently, Zhou and colleagues [88] employed a bioinformatics approach and harnessed the SEMA6C gene expression profile, among 11 other chosen genes, to build a prediction algorithm for the immunological microenvironment of pancreatic tumors. According to the authors’ findings, the proposed method based on the immune-related gene prognostic index (IRGPI) can be employed in the survival prediction (OS) of patients with pancreatic adenocarcinoma (PAAD) and molecular classification of PAAD.

Lines 487-489 - Interestingly, the authors of the study concluded that SEMA6D may serve as a sensitive marker of liver injury in CHC (superior to APRI and FIB-4 in predicting the development of cirrhosis) [105].

Lines 506-509 - In other words, upregulated SEMA6D levels in cisplatin-resistant osteosarcoma cells may be used as a prediction tool for personalized therapy and considered a promising target for the molecular targeted therapy of chemoresistant osteosarcoma [118].

Lines 559-562 - Taken together, this study emphasized the crucial role of SEMA6D/plexin-A1 signaling activity in the survival pathway of human mesothelioma cells and underlined a possible molecular target for therapeutic modalities.

Lines 571-573 - Taken together, these reports highlight the substantial role of SEMA6D in the tumor microenvironment and suggest the SEMA6D/VEGF(R) axis as a possible target for pharmacological intervention.

Lines 583-585 In terms of these results, the authors concluded that SEMA6D may serve as a reliable biomarker to track ccRCC development.

Reviewer 3 Report (New Reviewer)

Comments and Suggestions for Authors

The review by Wagner et al. “Semaphorin 6 family - an important yet overlooked group of signaling proteins involved in cancerogenesis” summarizes important information about the role of semaphorins in health and disease, in particular in cancer pathology. The review is very well illustrated, which contributes to the understanding of the complex intracellular signaling pathways in which proteins of the SEMA6 family are involved.

Reviews devoted to one gene encoding a regulatory protein or a group of similar genes always arouse keen interest among readers. Young scientists find in them new directions and justification for their research, and pharmaceutical industry specialists find a theoretical basis for the development of new therapies.

There are several minor comments to the text of the review.

Table 1. Is Table 1 visible in total in the entire file? If it is visible in the file in full, then it can be made more compact by removing empty lines. I counted only 8 lines, including the table header, that contain information.

Line 547. Specify which semaphorin you are talking about. Written SEMAD, probably SEMA6D.

Comments on the Quality of English Language

Moderate English editing required. The main focus should be on using the correct forms of verbs and introductory words.

Author Response

Reviewer 3: The review by Wagner et al. “Semaphorin 6 family - an important yet overlooked group of signaling proteins involved in cancerogenesis” summarizes important information about the role of semaphorins in health and disease, in particular in cancer pathology. The review is very well illustrated, which contributes to the understanding of the complex intracellular signaling pathways in which proteins of the SEMA6 family are involved.

Reviews devoted to one gene encoding a regulatory protein or a group of similar genes always arouse keen interest among readers. Young scientists find in them new directions and justification for their research, and pharmaceutical industry specialists find a theoretical basis for the development of new therapies.

There are several minor comments to the text of the review.

Comment 1.  Table 1. Is Table 1 visible in total in the entire file? If it is visible in the file in full, then it can be made more compact by removing empty lines. I counted only 8 lines, including the table header, that contain information.

Response to Comment 1. Thank you for pointing to this issue. Table 1 has been made more compact according to your suggestion. Reports on tumors and associated semaphorin 6 expression have been arranged in the table according to origin of malignant tissue.

Comment 2.  Line 547. Specify which semaphorin you are talking about. Written SEMAD, probably SEMA6D.

Response to Comment 2. Thank you for pointing to this issue. Text line 548 has been corrected

“According to Lu and colleagues, the SEMA6D-plexin-A1 complex was highly expressed in vascular endothelial cells within gastric cancer and positively correlated with VEGFR2.“

Comment 3.  Moderate English editing required. The main focus should be on using the correct forms of verbs and introductory words.

Response to Comment 3. Thank you for pointing to this issue. The final version of the manuscript has been checked by a colleague fluent in English writing followed by language editing with American Journal Experts Digital Editing tool.

This manuscript is a resubmission of an earlier submission. The following is a list of the peer review reports and author responses from that submission.

Round 1

Reviewer 1 Report

Comments and Suggestions for Authors

After a detailed analysis of the manuscript entitled “Semaphorin 6 family - an important yet overlooked 2 group of signaling proteins involved in cancerogenesis”. I have some major corrections that need to be incorporated in the manuscript.

1. The manuscript has not presented a very detailed presentation of the literature. The paragraphs are not constructed properly and an in depth detail of the topics seems lacking.

2. The introduction section should be re-written properly by including detailed presentation of the Semaphorin-6 family along with the other semaphorin family members such as semaphorins 3,4,5 and 7.

3. Association of the semaphorin 6  family with other proteins and targets should also be included in the manuscript.

4. The manuscript only has a single figure; However, atlease 3-4 figures explaining the signaling pathways and semaphoring 6 family must be included.

5. The role of semaphorin 6 family in carcinogenesis has been explained but it seems very confusing.

Comments on the Quality of English Language

Fair

Reviewer 2 Report

Comments and Suggestions for Authors

In this review manuscript, Wagner et al updated the literature relevant to the involvement of the Sema6 family (Sema6A-6D) in oncogenesis.

This review article will attract the cancer research community’s attention to the contributions of Sema6 members to a variety of aspects of cancer, including progression, biomarker, therapeutic targets, and tumor microenvironment with respect to immune alterations. The manuscript is clearly structured, easy to read, and contributes the current literature.

Nonetheless, some revisions are suggested for improvement.

There is a relatively large size of publications related to Semaphorin and cancer in PubMed. The authors seem to suggest the topic was largely understudied. It will be helpful to indicate the criteria used to select articles and the number of publications being reviewed.

Line 94, please revise “cancer tumors”.

Line 148, for refs 31 and 32, please describe these studies in more detail to support the message here.

Line 257, please cite the reference for the study by Li et al.

In the line-248 paragraph, authors seem to indicate the downregulation of SEMA6B in breast cancer due to methylation alterations. In the line-275 paragraph, MCF7 cells express a high level of SEMA6B. Please clarify it.

For the line-342 paragraph, can a figure be prepared to illustrate its content?

Based on the content of the line-405 paragraph, the oncogenic impact of Sema6D remains unclear or is content-dependent. Nonetheless, the authors summarized it as “… … SEMA6D in maintaining the balance of tumorigenicity in breast cancer”. Tumorigenesis is resulted from process unbalance; “balance of tumorigenicity” is hard to understand. Please revise it to meet the content of the paragraph.

Both Sema6D and SEMA6D were used. Please use either style, but not both.

Lines 405 and 438, should SemaD be Sema6D?

The “Concluding Remarks” is too general; please revise it to strength its cancer relevance. 
